# Reconstructing Snow Cover under Clouds and Cloud Shadows by Combining Sentinel-2 and Landsat 8 Images in a Mountainous Region

**Yanli Zhang** [1,2,*] **, Changqing Ye** [1] **, Ruirui Yang** [1] **and Kegong Li** [3]

1   College of Geography and Environment Science, Northwest Normal University, Lanzhou 730070, China; 2021212870@nwnu.edu.cn (C.Y.); 2022213001@nwnu.edu.cn (R.Y.)
2   Key Laboratory of Resource Environment and Sustainable Development of Oasis, Lanzhou 730070, China
3   Gansu Provincial Institute of Surveying and Mapping Engineering, Lanzhou 730030, China; likegong513@126.com
*   Correspondence: zyl0322@nwnu.edu.cn

**Abstract:** Snow cover is a sensitive indicator of global climate change, and optical images are an important means for monitoring its spatiotemporal changes. Due to the high reflectivity, rapid change, and intense spatial heterogeneity of mountainous snow cover, Sentinel-2 (S2) and Landsat 8 (L8) satellite imagery with both high spatial resolution and spectral resolution have become major data sources. However, optical sensors are more susceptible to cloud cover, and the two satellite images have significant spectral differences, making it challenging to obtain snow cover beneath clouds and cloud shadows (CCSs). Based on our previously published approach for snow reconstruction on S2 images using the Google Earth Engine (GEE), this study introduces two main innovations to reconstruct snow cover: (1) combining S2 and L8 images and choosing different CCS detection methods, and (2) improving the cloud shadow detection algorithm by considering land cover types, thus further improving the mountainous-snow-monitoring ability. The Babao River Basin of the Qilian Mountains in China is chosen as the study area; 399 scenes of S2 and 35 scenes of L8 are selected to analyze the spatiotemporal variations of snow cover from September 2019 to August 2022 in GEE. The results indicate that the snow reconstruction accuracies of both images are relatively high, and the overall accuracies for S2 and L8 are 80.74% and 88.81%, respectively. According to the time-series analysis of three hydrological years, it is found that there is a marked difference in the spatial distribution of snow cover in different hydrological years within the basin, with fluctuations observed overall.

**Keywords:** Sen2Cor; cloud detection; SNOWL; Fmask4.0; Babao River Basin

## 1. Introduction

Snow cover is one of the most active land cover types on the Earth's surface. It has high albedo and low thermal conductivity, which strongly influence the global water cycle, energy balance, and climate change [1–5]. Optical remote-sensing satellites are important data sources for snow cover detection, and the normalized difference snow index (NDSI) proposed by Dozier [6] can effectively identify snow cover through its high reflectivity in the green band (0.5 μm) and strong absorptivity in the shortwave infrared band (1.6 μm). Recently, both geostationary and polar-orbiting satellites released various snow cover products at no cost, including Meteosat/MSG, MODIS, AVHRR, and FY [7–9]. However, snow cover in mountainous areas has the characteristics of high albedo, strong spatial heterogeneity, and a fast change rate. Therefore, the snow cover products mentioned above are limited: a coarser spatial resolution cannot describe the spatiotemporal distribution characteristics of snow cover in detail, while a lower radiometric resolution leads to saturation for snow [10–12].

In recent years, the successful launch of the Sentinel-2 (S2) and Landsat 8 (L8) satellites has brought new opportunities for mountainous snow cover identification and detection. Compared to the Landsat 1–7 series satellites, L8 has a 12-bit radiometric resolution and reduced saturation for snow. However, due to its low revisit period (16 days), using L8 alone cannot capture the rapid changes in mountainous snow cover. The S2 satellite not only provides higher spectral resolution, radiometric resolution, and spatial resolution, but its A/B satellite network can also provide two to five days of higher temporal resolution images [13]. Therefore, the combination of S2 and L8 imagery is beneficial for studying the rapid changes and redistribution process of mountainous snow cover.

However, optical satellites are severely influenced by clouds and cloud shadows (CCSs), and nearly half of multispectral images are polluted [14,15]. Although most of the snow and clouds can be distinguished by the NDSI, it cannot recover snow cover under CCSs. Clouds alter the energy radiation transmission between sun-surface sensors [15–17], making it difficult for snow information under clouds to accurately reach sensors. Moreover, the spectral characteristics of its shadows are relatively similar to those of wetlands, water, and other ground objects, thereby reducing the recognition accuracy of snow cover beneath cloud shadows [18,19]. Generally, the spectral reflectance of snow beneath thin cloud shadows is relatively high, and snow cover can be directly extracted by the NDSI. However, snow information under thick cloud shadows is difficult to identify. Therefore, how to combine S2 and L8 images to restore snow cover under CCSs is the primary necessary mission of long-term snow cover detection.

Based on MODIS snow cover products released globally, many studies have been conducted on recovering snow cover under clouds, mostly including temporal/spatial filtering, multisensor synthesis, and the SNOWL (snow line) algorithm using the digital elevation model (DEM) [20–25]. The temporal-filtering method utilizes the short-term dynamic change in cloud layers to recover snow cover beneath the clouds, but it usually requires very high temporal resolution satellite images, such as revisiting cycles of more than two times in a day. The spatial filtering method typically uses adjacent cloud-free pixel information, but it ignores rapid changes in snow cover and is not suitable for mountainous regions with significant spatial heterogeneity [26]. The multisensor synthesis method integrates optical satellite images with microwave satellite images or ground observation data to enhance the recognition capability of snow cover under clouds [25,27]. However, the effectiveness of this method is limited by the low spatial resolution of passive microwave remote-sensing data and the sparsity of ground observation stations in mountainous regions. Compared with the above three methods, the SNOWL developed by Parajka et al. [21] is more suitable for S2 and L8 images with a lower temporal resolution. By comparing the elevation of cloud pixels and the regional average snow line elevation, cloud pixels are redefined as land pixels or snow cover. In addition, in previous MODIS snow products, the focus was on the impact of clouds on snow cover, neglecting the extraction of snow cover under cloud shadows. However, for high-resolution images, cloud shadows are also an important factor in extracting snow cover.

Obviously, for S2 and L8 images with a low temporal resolution and high spatial resolution, the SNOWL algorithm has become the main method for snow cover reconstruction under CCSs, and cloud and cloud shadow detection is a prerequisite. There are two main types of algorithms used for CCSs detection: machine learning and spectral threshold detection. Machine-learning methods require many training samples and have difficulty achieving global universality [28–30]. Spectral threshold detection methods utilize the special physical features of clouds, such as brightness, whiteness, and coolness, to construct multiple spectral indices, which are widely used [12,31–40]. For the problem of cloud pollution in Landsat-series satellite, some scholars have proposed various CCSs algorithms based on spectral threshold detection methods, such as ACCA, Fmask, MCM, and LaSRC, through the differences in surface reflectance and temperature between clouds and ground objects [31,34–36]. The ACCA cloud detection algorithm targets Landsat 4/5 and Landsat 7 images, with high cloud omission and misalignment errors, and

does not detect cloud shadows [32,33]. For high-resolution images, the impact of cloud shadows on snow reconstruction cannot be ignored. The Fmask algorithm was first used for Landsat 7 images [34], which have poor recognition accuracy for thin cirrus clouds, and the improved Fmask 3.3 algorithm greatly improved its detection accuracy by utilizing the cirrus band of Landsat 8 [19]. Candra et al. [35] developed the MCM algorithm for Landsat 8 to detect CCSs through the difference in reflectivity between cloud-covered and clear-sky pixels, but this is only effective for thick clouds and their shadows. LaSRC generates the cloud mask of L8 images during atmospheric correction and extracts cloud shadows by band thresholds, but the detection accuracy of thin clouds is low [41]. The above research indicates that the Fmask algorithm and its higher version can effectively detect CCSs information in Landsat-series satellite images.

With the wide use of Sentinel-2 A/B images, many researchers have proposed CCSs detection algorithms [12,37–40,42,43]. The Sen2Cor tool developed by ESA can effectively distinguish snow cover, clouds, and cloud shadows on S2 images [42]. Because of the absence of brightness temperature (BT) observed in a cloud-sensitive thermal infrared band in S2 images, some scholars have tried to improve Landsat CCSs detection algorithms and apply them to Sentinel-2 images. Among them, MAJA is extensively applied in Landsat and Sentinel-2 images but overestimates CCSs [39,40,43]. Zhu and Helmer [37] developed the ATSA algorithm by utilizing the spectral characteristics of clouds and the geometric relationships between clouds and their shadows. This algorithm was used in Landsat 4/8 and Sentinel-2 images; the separation effect of clouds and snow clouds and snow was poor. Qiu et al. [38] designed a cloud probability based on haze optimization transformation (HOT) to replace the temperature probability of S2 images and proposed the Fmask4.0 algorithm suitable for both Landsat-series satellites and Sentinel-2. Candra et al. [40] improved the MCM algorithm by using the similarity of spectral features of S2 and L8 images and used multitemporal images to optimize CCS detection capabilities; however, the improved algorithm has strict requirements for clear-sky reference images. Tarrioa et al. [43] conducted a comparative analysis of five cloud detection algorithms (MAJA, LaSRC, Tmask, Sen2Cor, and Fmask4.0) applied to Sentinel-2 images and found that the Fmask4.0 algorithm has high accuracy and fast operation speed. However, its cloud shadow detection is determined by the geometric position relationships among CCSs, which can easily mix with dark surface pixels (water, hillshade, etc.), resulting in cloud shadow omission. In our published study [12], the Fmask4.0 algorithm is improved based on the darker characteristics of cloud shadows in the near-infrared/shortwave infrared (NIR/SWIR1) bands, but there is still a problem of cloud shadow underestimation.

After the detection of CCSs, the SNOWL algorithm is applied to redefine cloud and cloud shadow pixels with elevations greater than the average snow line as snow cover under CCSs. Obviously, clear-sky snow cover extraction is a prerequisite for the SNOWL algorithm to determine the instantaneous snow line. According to the high reflectivity of snow, a comprehensive combination of various thresholds, such as the NDSI, visible light, and NIR bands, can easily distinguish snow from other land covers. Numerous studies have adopted nearly the same clear-sky snow cover extraction method [6,44–47]. In our previous study, the Sen2Cor tool was able to extract Sentinel-2 images of clear-sky snowpack pixels with high precision. Nevertheless, the original SNOWL method has some shortcomings, such as not being able to effectively detect CCSs or not considering the impact of unstable snowpack areas when estimating instantaneous snow lines [48–50]. Gascoin et al. [51] used the LIS (let-it-snow) open-source processor to extract clear-sky snow based on the NDSI and red bands of S2 and L8 images, and then reconstructed the snow cover under clouds using the snow line elevation, but could not detect thick cirrus clouds. Premier et al. [52] used images fused by MODIS and Sentinel-2/Landsat 8 and comprehensively considered historical snow cover information (snow condition probability) to reconstruct snow cover under cloud-covered images, which relies heavily on the accuracy of historical snow cover information. Our previous study [12] improved the SNOWL algorithm by introducing unstable snow cover areas (USCAs), which improved the reconstruction accuracy of snow

cover on S2 images. However, the cloud shadow recognition effect was not ideal, and the improved algorithm has not been applied to Landsat 8 satellites.

The purpose of this study is to improve the Fmask4.0 cloud shadow detection algorithm and implement different cloud detection methods for L8 and S2 images for reconstructing snow cover from September 2019 to August 2022 in the Babao River Basin based on the GEE platform. The paper is organized as follows: Section 2 introduces the study area and datasets. Section 3 outlines the snow cover reconstruction methods for S2 and L8 images. Sections 4 and 5 present the results and discussion. Finally, Section 6 concludes the paper.

## 2. Study Area and Data

### 2.1. Study Area

The Babao River Basin (BRB), situated in the northeast of the Qilian Mountains in China, has approximately 105 km in length and a total area of around 2452 km² [53], as displayed in Figure 1. The watershed exhibits a complex terrain, surrounded by high mountains, with elevations ranging from 2673 to 4960 m. The dominant land cover types are grassland (77.20%), bare land (11.14%), moss (8.61%), and alpine woodland (1.76%) [54,55]. The average annual temperature is −4.2 °C [54], which has the characteristics of a continental climate and plateau mountain climate [56]. It has an extensive range of snow cover and frequent snowfall in winter, mainly from October to May of the subsequent year [57]. There are important snow observation sites within the watershed that are convenient for snow monitoring, such as the Dadongshuyakou Observation Station. Due to its unique geographical location and hydrological characteristics, the BRB has become a desired area for researching snow cover variation in cold regions.

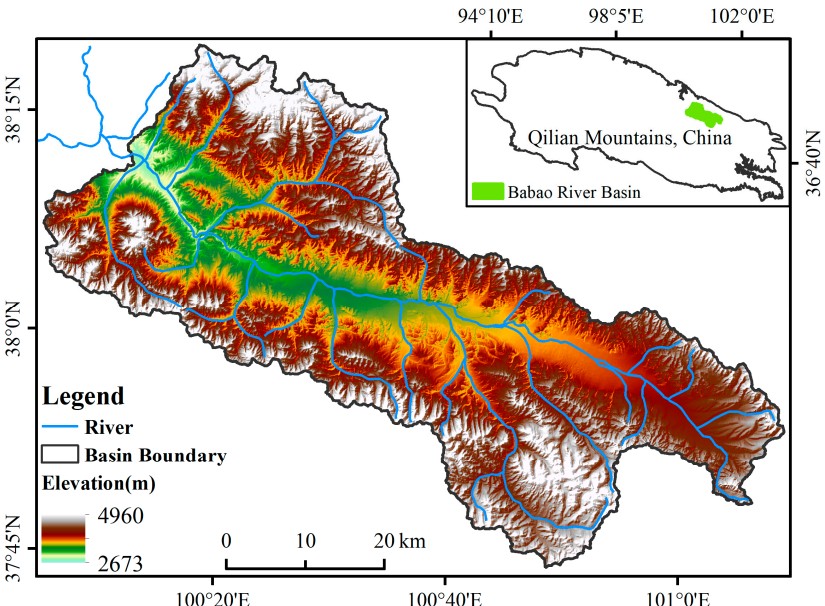

**Figure 1.** Location of the study area.

### 2.2. Datasets

The data used in the paper mainly include two categories, Sentinel-2 A/B and Landsat 8 multispectral images, and SRTM DEM, ERA5-Land Daily, GF-2, and other auxiliary data, as shown in Table 1. Multispectral data, SRTM DEM, and ERA5-Land Daily are obtained from the GEE datasets. GF-2 is used for accuracy verification from the State Key Laboratory of Remote Sensing Science.

**Table 1.** Basic information of experimental datasets.

| Datasets | | Track/Position Information | Spatial/Radiometric Resolution | Data Period | Scenes | Application |
|---|---|---|---|---|---|---|
| **Multispectral images** | Sentinel-2 L1C/L2A | 47SPC, 47SPB, 47SPC | 10, 20, 60 m, 12 bits | 2019/09/01–2022/08/31 | 399 | Snow cover extraction |
| | Landsat 8 TOA/SR | Path 133/Row 34 | 30 m, 12 bits | | 35 | |
| **Auxiliary** | SRTM DEM | Babao River Basin | 30 m | — | — | Aspect and snow line height extraction |
| | ERA5-Land Daily | | 0.1° | 2019/09/01–2022/08/31 | — | Air temperature |
| | GF-2 | 100.8°E _37.9° N 100.5°E _37.9°N | 0.8 m | 2020/03/23 2020/01/14 | 2 | Accuracy verification |

### 2.2.1. Multispectral Images

This article uses S2 and L8 images to extract snow cover information. Except for the Sentinel-2 non-BT band, the difference in visible and near-infrared bands between the two images is very small [58,59]. Sentinel-2 L2A is a surface reflectance product obtained by atmospheric correction of the atmospheric apparent reflectance (TOA) product (L1C). Due to the lack of the cirrus band required for CCSs detection, it is necessary to introduce the cirrus band from L1C data. Landsat 8 SR is surface reflectance data. Similarly, cloud detection is conducted by the cirrus band. To perform CCSs identification, all data were resampled to a spatial resolution of 10 m. The entire BRB requires three scenes of Sentinel-2 images or one scene of a Landsat 8 image to cover. A total of 399 Sentinel-2 images with 133 days and Landsat 8 images with 35 days were obtained from September 2019 to August 2022.

### 2.2.2. Auxiliary Data

The auxiliary data mainly include DEM, ERA5-Land Daily, and GF-2 images. The DEM comes from the Space Shuttle Radar Mission (SRTM), which collected over 80% of surface digital terrain information between 60°N and 56°S [60], with a spatial resolution resampling to 10 m to obtain snow line height and aspect. ERA5-Land Daily obtained from GEE provides daily average air temperatures at 2 m above the land surface. Two scenes of GF-2 images with a 0.8 m spatial resolution are applied to verify the extraction results of cloud-free snow cover and snow cover reconstruction under CCSs. Due to the fact that GF-2 lacks the near-infrared feature band (1.6 um) for extracting snow, the *MeanVis* method proposed by Zhang et al. [12] can improve the underestimation phenomenon of snow cover in mountainous shadow regions when extracting snow cover using GF-2.

### 3. Methodology

This study used JavaScript language to process S2 and L8 data on the GEE platform, and modified the Python version of Fmask4.0 algorithm to a JavaScript version supported by GEE. Additionally, both Sen2Cor and SNOWL algorithms were also written in JavaScript. As shown in Figure 2, snow cover reconstruction under CCSs for S2 and L8 consists of three steps: detecting CCSs according to the improved Fmask4.0, extracting cloud-free snow cover pixels using Sen2Cor, and reconstructing snow cover under CCSs according to our proposed improved SNOWL algorithm that introduces unstable snow cover areas.

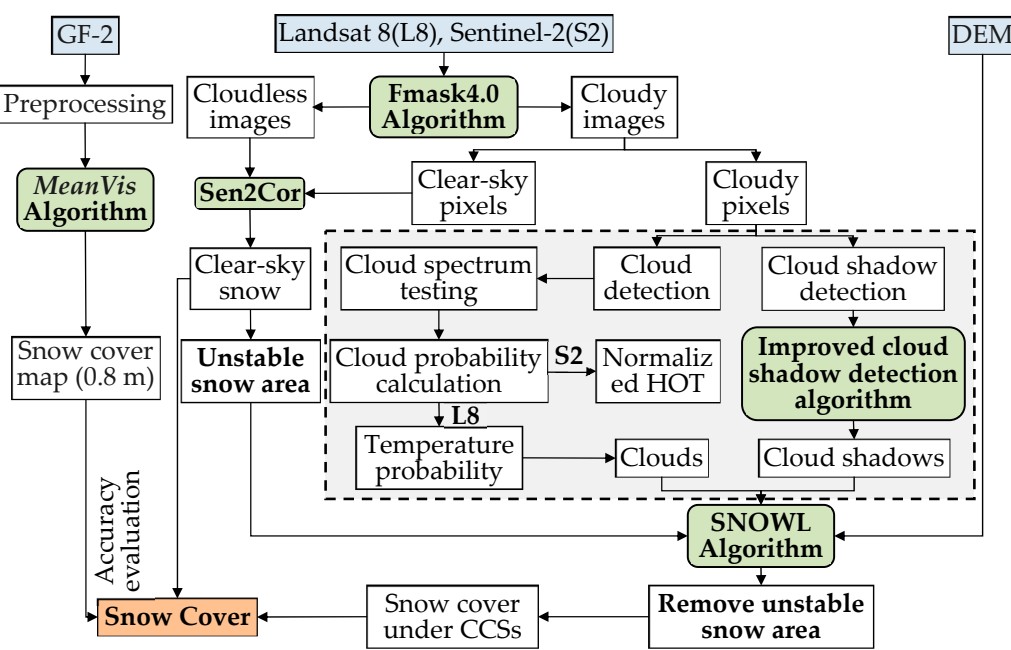

**Figure 2.** Flowchart of this algorithm.

### *3.1. Cloud and Cloud Shadow Detection*

#### 3.1.1. Cloud Detection

The key technology for distinguishing clouds from surface objects is cloud probability calculation and cloud spectrum testing, which are mainly based on the brightness, whiteness, and coldness characteristics of clouds. Cloud spectrum testing is used to obtain potential cloud pixels through five steps, Basic Test, Whiteness Test, HOT Test, Rock Test, and Cirrus Test, and the specific formulae are detailed in the literature of Zhu et al. [19]. Different methods were selected based on S2 and L8 images in the Basic Test because S2 does not provide a thermal infrared band for capturing the "coldness" characteristic of clouds, while other test modules are universal for both types of images. For Landsat 8, cloud pixels can be detected when the reflectivity value of the SWIR2 band is greater than 0.03 and the value of BT is less than 27 °C, and the values of the NDVI and NDSI are both less than 0.8, as shown in Formula (1). For Sentinel-2, cloud detection relies on the NDSI, NDVI, and SWIR2 band thresholds, which are consistent with Formula (1), except for BT.

$$L_{Basic} = \rho_{SWIR2} > 0.03 \text{ and } BT < 27 \,°\text{C and } NDSI < 0.8 \text{ and } NDVI < 0.8 \tag{1}$$

$$S_{Basic} = \rho_{SWIR2} > 0.03 \text{ and } NDSI < 0.8 \text{ and } NDVI < 0.8 \tag{2}$$

where $L_{Basic}$ and $S_{Basic}$ represent the Basic Test of Landsat 8 and Sentinel-2, respectively. $\rho_{SWIR2}$ is reflectivity value of the SWIR2 band, and $BT$ represents the brightness temperature band of Landsat 8.

Through the above five cloud spectrum tests, most of the thick clouds are effectively recognized. However, some thin clouds, clouds located at the edges, and some fragmented clouds are still difficult to extract. Therefore, it is necessary to introduce the cloud probability to further detect these cloud pixels. Due to the lack of large-scale water in the BRB, this study only focuses on calculating the probability of clouds over land. Generally, cloud probability is determined by the combination of spectral change rate, temperature probability, and the Cirrus Test. Similarly, due to the absence of the temperature band in Sentinel-2, different cloud probability formulae are used for the two satellite datasets, as shown in Table 2.

**Table 2.** Cloud probability formulae for S2 and L8.

| | Sentinel-2 | Landsat 8 | Equation Number |
|---|---|---|---|
| **Spectral change rate** | $l_{vari} = 1 - \max(White, abs(NDVI), abs(NDSI))$ | | (3) |
| **Temperature probability** | $\begin{cases} HOT = (\rho_{blue} - 0.5 \times \rho_{red} - 0.08) \\ l_{HOT} = \dfrac{HOT - HOT_{low} - 0.04)}{(HOT_{high} + 0.04) - (HOT_{low} - 0.04)} \end{cases}$ | $l_{temper} = \dfrac{(T_{high} + 4 - T_{BT})}{(T_{high} + 4 - (T_{low} - 4))}$ | (4) |
| **Cloud probability** | $SCloud_P = (l_{vari} \times l_{HOT} + l_{Cirrus} \times 0.5) > 0.8$ | $LCloud_P = (l_{vari} \times l_{temper} + l_{Cirrus} \times 0.3) > 0.8$ | (5) |

where *White*, $l_{vari}$, and $l_{Cirrus}$ are the Whiteness Test, spectral change rate, and cirrus probability for L8 and S2, respectively. *HOT*, $l_{HOT}$, and $SCloud_p$ represent the HOT Test, the temperature probability, and cloud probability for Sentinel-2, respectively. $l_{temper}$ and $LCloud_p$ are the temperature probability and cloud probability for Landsat 8, respectively. The detailed methods and terminologies applied in the formulae in Table 2 can be found in Qiu et al. [38].

### 3.1.2. Improved Cloud Shadow Detection

Due to the low reflectivity of cloud shadows in both the NIR and SWIR1 bands, the original Fmask algorithm extracted cloud shadows using threshold methods with reflectivities less than 0.25 and 0.11 in the two bands, respectively [61]. However, the cloud shadow detection algorithm in Fmask4.0 has two shortcomings: first, it is difficult to distinguish between water and cloud shadows; second, when cloud shadows fall on bright surface areas, such as bare areas, they are usually unrecognizable. However, experiments have found that this method can effectively identify large homogeneous cloud shadows, and cloud shadow pixels located at their boundaries are usually overlooked. Furthermore, the reflectivity of cloud shadows is influenced and is also difficult to distinguish from that of water. Therefore, an improved cloud shadow detection algorithm is proposed by removing water and considering whether the surface is covered by vegetation. The thresholds of NIR, SWIR1 band, and *NDWI* were obtained by many experiments on cloud shadows in the BRB, as shown in Formulae (6) and (7).

$$NDWI < 0.1 \tag{6}$$

$$CS = \begin{cases} \rho_{NIR} < 0.25 \text{ and } \rho_{SWIR1} < 0.11, & NDVI > 0.08 \\ \rho_{NIR} < 0.3 \text{ and } \rho_{SWIR1} < 0.28, & NDVI \leq 0.08 \end{cases} \tag{7}$$

where CS represents cloud shadow.

### 3.2. Snow Cover Extraction

Snow extraction consists of two parts based on whether the surface is covered by CCSs. First, the Sen2Cor algorithm provided by ESA was used to extract the cloud-free snow cover. Subsequently, for pixels covered by CCSs, the published improved SNOWL algorithm is adopted to reconstruct snow cover under CCSs.

### 3.2.1. Cloud-Free Snow Cover Extraction

Sen2Cor provides an algorithm for snow cover identification in Sentinel-2 images. It uses four thresholds (NDSI, blue band, blue-green ratio, and NIR band) to extract cloud-free snow cover pixels in S2 images. Because L8 provides five spectral bands similar to S2 (blue, green, red, NIR, and SWIR1), the same algorithm is used for snow extraction of cloud-free pixels. The specific formulae can be found in our published article [12].

### 3.2.2. Snow Cover Extraction under CCSs

For high-resolution Landsat 8 and Sentinel-2 satellites with longer revisit periods, the SNOWL algorithm is generally selected to recover snow cover under CCSs based on the snow line elevation. However, the ablation and accumulation of mountainous snow cover are related to the surface elevation and are affected by the aspect, surface irradiance,

and other factors. Therefore, in our published research, the original SNOWL algorithm is modified by introducing unstable snow cover areas to improve the accuracy of snow cover reconstruction [12].

To recover snow cover under CCSs on S2 and L8, in this study, the improved SNOWL algorithm is used, which involves three steps: (1) Based on clear-sky snow cover of two satellite images over three hydrological years, pixels with less than 22 days of snow cover within a year were identified as USCA. (2) We use the original SNOWL algorithm by comparing the elevation of CCSs with the average snowline elevation of each scene image to identify the initial snow cover pixels. (3) Combining the above two results, the initial snow pixels located in the USCA are eliminated. Thus, the snow cover pixels under CCSs are obtained. In this paper, the USCA is extracted from all available cloud-free snow cover pixels of S2 and L8.

## 4. Results

### 4.1. Evaluation of Cloud-Free Snow Cover

The extraction of cloud-free snow cover on Sentinel-2 and Landsat 8 images is the foundation for the reconstruction of snow cover under CCSs. In the paper, the snow recognition results of GF-2 satellite images (0.8 m) are considered the "true value" to verify the ability of the Sen2Cor algorithm to detect cloud-free snow cover on S2 and L8 images. Due to data limitations, the GF-2 and L8 images used for validation were acquired on 14 January 2020 but differed by one day from S2 (13 January 2020). Five evaluation indicators are selected: overall accuracy $O$, multimeasurement error $M$, omission error $L$, user accuracy $U$, and *Kappa* coefficient. The unit for the first four indicators is percentage (%). The detailed definitions and calculation formulae are detailed in our published literature [12].

Figure 3 illustrates that the spatial distribution of clear-sky snow cover extracted from two satellite images is very similar to that of GF-2. Detailed accuracy results are indicated in Table 3. The overall accuracies of the S2 and L8 images are 84.51% and 80.40%, respectively, suggesting that the Sen2Cor algorithm can properly classify over 80% of the pixels in the two satellite images. From the accuracy indicators listed in Table 3, it can be observed that the extraction accuracy of snow cover on S2 is generally higher than that of L8. However, the multimeasurement error is poor (43.19%) for the transit time difference between Sentinel-2 and GF-2, although its user accuracy is very high (93.77%).

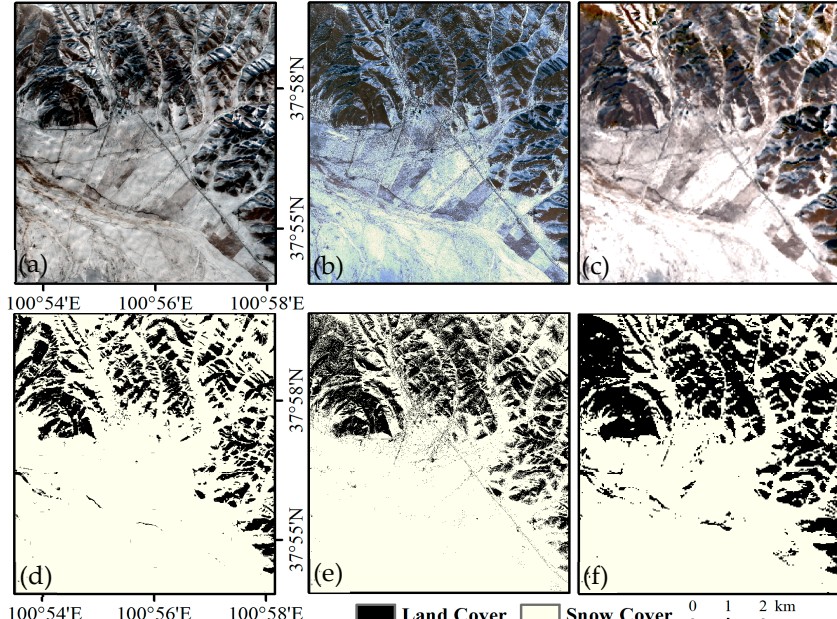

**Figure 3.** Comparative analysis of cloud-free snow cover detection using S2, L8, and GF-2 images. (**a**–**c**) Original images of Sentinel-2 on 13 January 2020, GF-2, and Landsat 8 on 14 January 2020; (**d**–**f**) Clear-sky snow cover of S2 on 13 January 2020, GF-2, and Landsat 8 on 14 January 2020.

**Table 3.** Detection accuracy of clear-sky snow cover on S2 and L8 images.

|  |  | Sentinel-2 (2020/01/13) | | Landsat 8 (2020/01/14) | |
|---|---|---|---|---|---|
|  |  | **Snow Pixel** | **Snow-Free Pixel** | **Snow Pixel** | **Snow-Free Pixel** |
| **GF-2 (2020/01/14)** | **Snow pixel** | 508,205 | 33,788 | 366,637 | 67,204 |
|  | **Snow-free pixel** | 78,259 | 102,959 | 52,578 | 124,745 |
| **Evaluating indicator** | | *U* | *M* | *L* | *O* |
| Sentinel-2 | | 93.77% | 43.19% | 6.23% | 84.51% |
| Landsat 8 | | 84.51% | 29.65% | 15.49% | 80.4% |

## 4.2. Evaluation of Cloud and Cloud Shadow Detection

To evaluate the detection accuracy of CCSs, S2 and L8 images with snow cover under CCSs were selected. Due to significant differences in the radiance values recorded in satellite images when cloud shadows fall on snow and snow-free covered surfaces, S2 and L8 images with snow and that are snow-free were selected to evaluate the cloud shadow detection effect. As depicted within the dotted box in Figure 4, the improved Fmask4.0 algorithm has a better recognition ability for cloud shadows on the four satellite images and obtains more detailed cloud shadows and their profile information (Figure 4c,g,k,o), while some cloud shadow pixels were not extracted in the original algorithm, as shown in the red circles in Figure 4d,h,l,p. Moreover, the Fmask4.0 algorithm also has high recognition accuracy for clouds in the four temporal images. It can detect uniform or edge clouds and effectively distinguish the pixels of clouds and snow. Notably, the highly reflective nature of snow cover allows snow cover to be extracted from thin cloud images directly using the Sen2Cor algorithm.

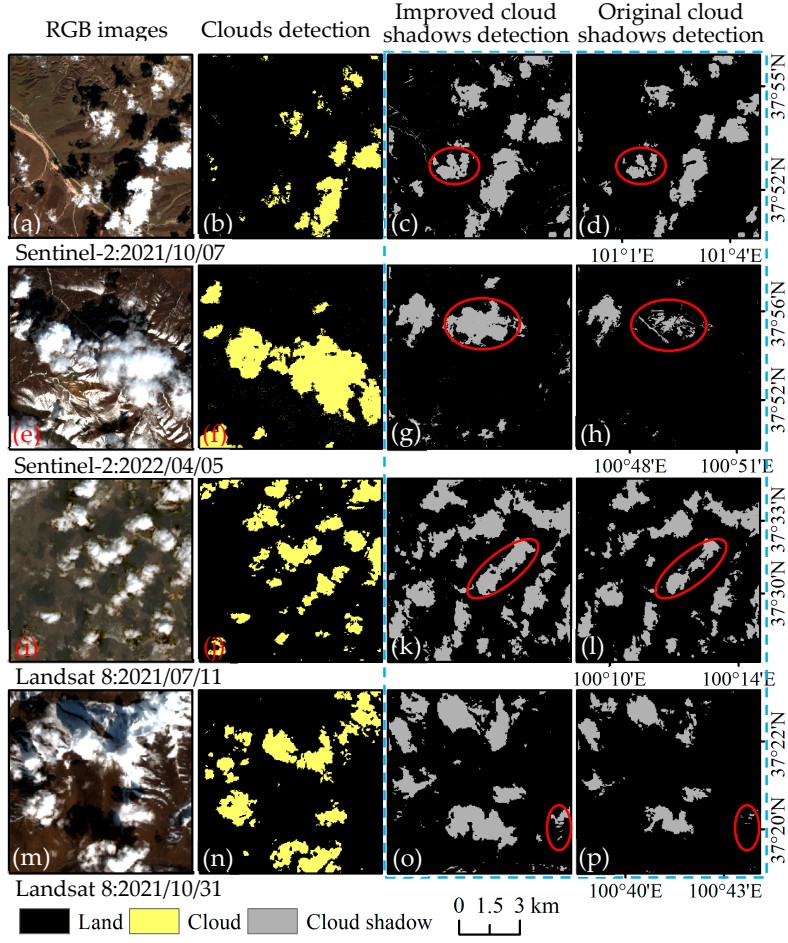

**Figure 4.** Comparison of CCS detection results between the improved Fmask4.0 and the original algorithm on S2 and L8 images: (**a**–**h**) S2 original image, clouds, and cloud shadows acquired by the

improved and original Fmask4.0 on snow-covered and snow-free surfaces; (**i–p**) L8 original image, clouds, and cloud shadows acquired by the improved and original Fmask4.0 on snow-covered and snow-free surfaces.

### 4.3. Snow Cover Reconstruction under CCSs

To assess the snow cover extraction results under CCSs, the Sentinel-2 image from 21 March 2020 was compared with the latest date GF-2 (23 March 2020). As shown in Figure 5, due to the inability to obtain cloud-free GF-2 data that meet the L8 transit date requirements, cross-validation was selected between S2 on 19 February 2021 and L8 on 17 February 2021. Historical temperature data provided by ERA5-Land Daily show that the temperature difference among the selected images is extremely small, and there is no obvious snowmelt. Therefore, the CCSs extracted from S2 and L8 are used as a mask to extract snow cover pixels from cloud-free GF-2 and S2 using this boundary to verify the reconstruction accuracy of snow cover beneath the CCSs.

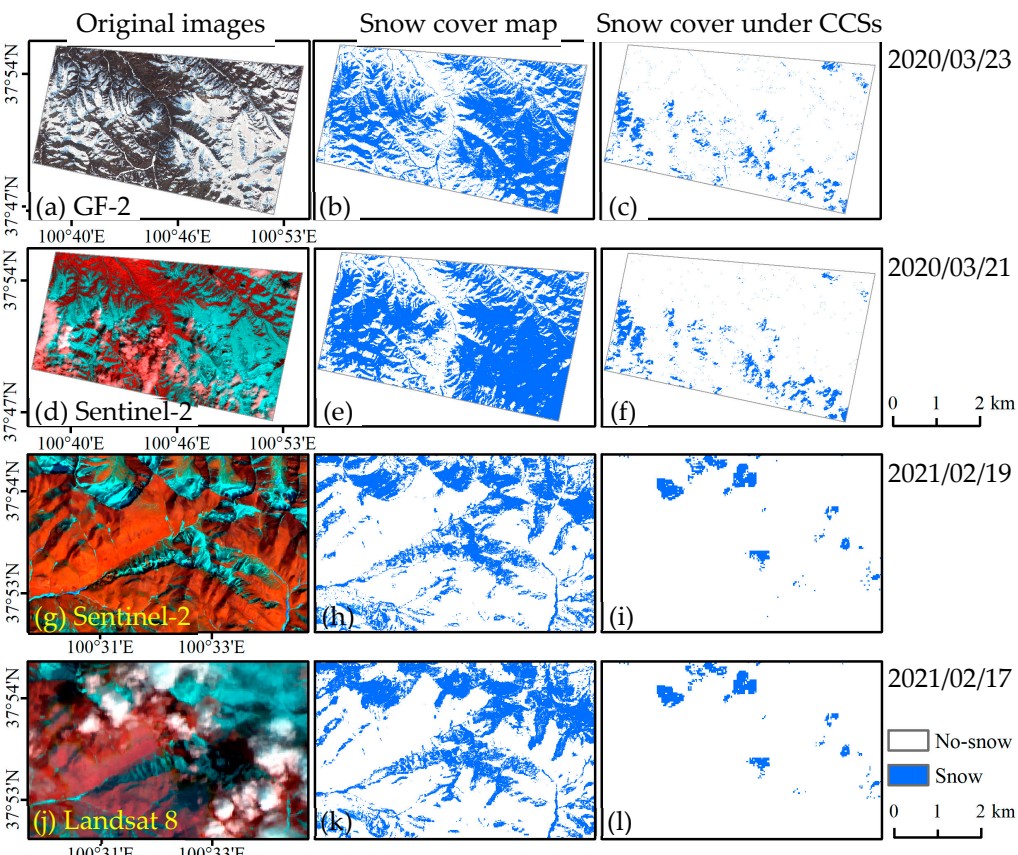

**Figure 5.** Accuracy evaluation of S2 and L8 snow cover reconstruction under CCSs: (**a–c**) The original image of GF-2, cloud-free snow cover, and snow cover under CCSs; (**d–f**) The original image of S2, cloud-free snow cover, and snow cover under CCSs obtained using the improved SNOWL algorithm; (**g–i**) The original image of S2, cloud-free snow cover, and snow cover under CCSs; (**j–l**) The original image of L8, cloud-free snow cover, and snow cover under CCSs obtained using the improved SNOWL algorithm.

Figure 5d,j describe the spatial distribution of CCSs in the false-color images of S2 and L8, and the reconstructed snow distribution under CCSs in both images (Figure 5f,l) is very similar to that of the corresponding validation data. By analyzing the confusion matrix of snow cover reconstruction results (Table 4), it was found that the overall accuracy of snow cover reconstruction on Sentinel-2 has increased from 77.27% to 80.74%. The user accuracy is 86.26%, and the Kappa coefficient is 0.616. Similarly, the corresponding accuracy of

Landsat 8 has also been improved, with 88.81%, 80.25%, and 0.766, respectively. The above experimental results indicate that our improved SNOWL algorithm can effectively recover mountainous snow cover under CCSs by introducing USCA. However, in areas with dense cloud cover (cloud coverage exceeding 30%), the reconstruction accuracy of snow cover is low, and the snow detail information is not good. Additionally, the effects of the nature of snow on the reconstruction results of snow cover under CCSs was not considered. The snow cover is mostly dry snow with high reflectivity and low and stable snow density in the BRB [55].

**Table 4.** Comparison of extraction accuracy of snow cover under CCSs on S2 and L8.

| | | Sentinel-2 (2020/03/21) | | | |
| | | Improved SNOWL | | Original SNOWL | |
| | | Snow Pixel | Snow-Free Pixel | Snow Pixel | Snow-Free Pixel |
|---|---|---|---|---|---|
| GF-2 (2020/03/23) | Snow pixel | 174,998 | 27,872 | 156,322 | 26,595 |
| | Snow-free pixel | 58,124 | 185,574 | 68,099 | 165,633 |
| | | Landsat 8 (2021/02/17) | | | |
| Sentinel-2 (2021/02/19) | Snow pixel | 11,941 | 2938 | 3430 | 1553 |
| | Snow-free pixel | 1239 | 21224 | 1510 | 10,390 |
| Evaluating indicator | | *U* | *M* | *L* | *O* | *Kappa* |
| S2 | Improved SNOWL | 86.26% | 23.85% | 13.74% | 80.74% | 0.616 |
| | Original SNOWL | 85.46% | 29.1% | 14.54% | 77.27% | 0.549 |
| L8 | Improved SNOWL | 80.25% | 5.52% | 19.75% | 88.81% | 0.766 |
| | Original SNOWL | 68.83% | 12.69% | 31.17% | 81.86% | 0.563 |

### 4.4. Impact of the Terrain on Accuracy of Snow Cover Reconstruction

Terrain is an important factor affecting the reconstruction of snow cover in mountainous areas, with elevation and aspect being the most significant. According to the difference in elevation across the study area, the elevation zone is carried out with a step length of 200 m; thus, the S2 and L8 images are divided into five elevation zones. According to the reconstruction results of snow cover in Section 4.3, the cloud-free snow cover pixels on GF-2 and Sentinel-2 are used as true values to evaluate Sentinel-2 on 21 March 2020 and Landsat 8 on 17 February 2021 in different elevation zones, as shown in Table 5. The findings indicate significant variation in the overall accuracy of snow cover reconstruction among different elevation zones for Sentinel-2, but it remains above 83%, and the overall accuracy is better than that of Landsat 8. Although Table 5 shows a decreasing trend in overall accuracy as elevation increases, this variation principle is unreliable due to the different SCRs in each elevation zone, especially at low altitudes.

**Table 5.** Impact of elevation on the reconstruction accuracy of snow cover.

| | Elevation | SCR | *U* | *O* | *M* | *L* |
|---|---|---|---|---|---|---|
| | Unit: m | | Unit: Percentage (%) | | | |
| S2 | 3474–3600 | 0.15 | 93.14 | 90.94 | 9.54 | 6.86 |
| | 3600–3800 | 4.49 | 88.90 | 88.06 | 12.11 | 11.10 |
| | 3800–4000 | 14.17 | 83.73 | 84.92 | 14.02 | 16.27 |
| | 4000–4200 | 18.43 | 88.63 | 86.98 | 15.26 | 11.37 |
| | 4200–4485 | 11.50 | 89.07 | 86.55 | 18.02 | 10.93 |
| L8 | 3327–3600 | 1.22 | 63.38 | 84.83 | 13.50 | 36.62 |
| | 3600–3800 | 4.57 | 64.73 | 85.81 | 5.37 | 35.27 |
| | 3800–4000 | 9.02 | 82.30 | 90.74 | 4.93 | 17.70 |
| | 4000–4200 | 13.08 | 89.65 | 91.84 | 5.85 | 10.35 |
| | 4200–4434 | 7.41 | 85.41 | 89.82 | 5.83 | 14.59 |

To analyze the impact of aspect on the accuracy of snow cover reconstruction, it is categorized into five groups: flat surface (−1), shady slope (0–45°, 315–360°), semishady slope (45–135°), sunny slope (135–225°), and semisunny slope (225–315°). As depicted in Table 6, the shady slope exhibits the highest overall accuracy for both S2 and L8, reaching 85.13% and 89.41%, respectively, followed by the semishady slope. In contrast, the overall accuracy on the sunny slope is the lowest because of the joint impact of the westerly wind, monsoon, and solar irradiance, which is 78.83% and 74.03% in S2 and L8, respectively. In addition, the SCR of aspects was not well-correlated with the reconstruction accuracy of snow cover in both satellite images.

**Table 6.** Impact of aspect on the reconstruction accuracy of snow cover.

| Aspect (°) | SCR | S2 U | O | M | L | SCR | L8 U | O | M | L |
|---|---|---|---|---|---|---|---|---|---|---|
| | | | | | Unit: Percentage(%) | | | | | |
| Flat surface | 10.81 | 81.83 | 79.96 | 21.67 | 18.17 | 6.18 | 82.74 | 87.78 | 7.10 | 17.26 |
| Shady slope | 6.79 | 84.95 | 85.13 | 14.26 | 12.05 | 10.53 | 82.20 | 89.41 | 5.68 | 17.80 |
| Semishady slope | 10.84 | 77.95 | 81.86 | 14.38 | 22.05 | 6.54 | 85.41 | 84.74 | 16.57 | 14.59 |
| Sunny slope | 7.56 | 71.93 | 78.83 | 19.37 | 28.07 | 3.96 | 73.72 | 74.03 | 25.69 | 26.28 |
| Semisunny slope | 12.72 | 76.12 | 79.28 | 17.22 | 23.80 | 11.08 | 79.61 | 77.96 | 22.40 | 20.39 |

*4.5. Mapping of Snow Cover*

In the paper, a total of 168 days of images from Sentinel-2 (133 days) and Landsat 8 (35 days) were collected for an analysis of snow cover changes in the BRB over three hydrological years from September 2019 to August 2022. Among them, 45 days of snow cover polluted by clouds was reconstructed using the improved SNOWL algorithm, including 39 days of Sentinel-2 and 6 days of Landsat 8. Figure 6 describes the temporal variation curve of the snow cover ratio (SCR) in the BRB for three hydrological years.

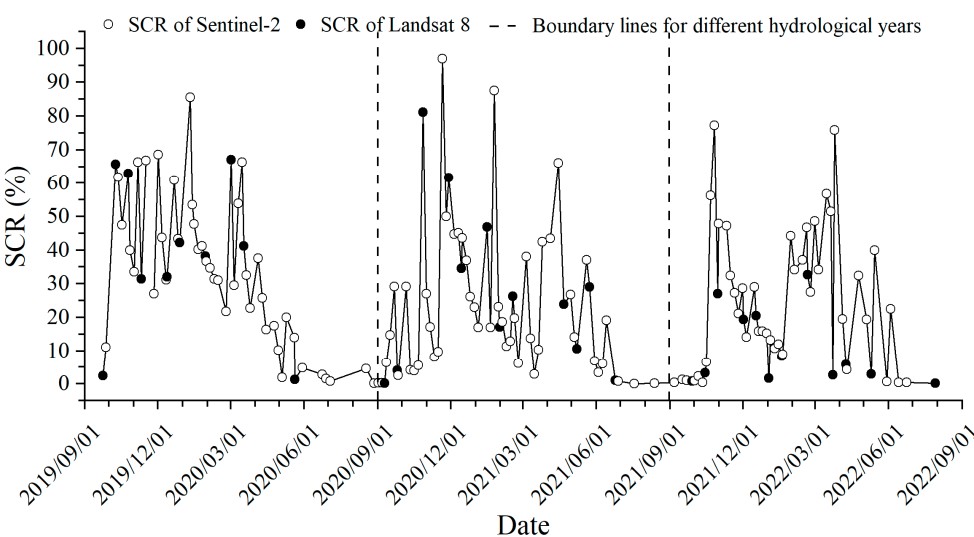

**Figure 6.** Variation characteristics of SCR of three hydrological years in the BRB.

In Figure 6, intra-annual and inter-annual variations in SCR are significant, and the curve fluctuates sharply with time. Generally, the snow cover area gradually increases from the end of September, and the distribution of snow cover throughout the winter is relatively widespread. However, the proportion of snow cover gradually diminishes as time progresses. Starting in spring, the SCR gradually increases and rapidly decreases after entering summer, even less than 1%. Furthermore, due to cloud interference and the limited temporal resolution, relying solely on Sentinel-2 images cannot satisfy the demand

for long-term snow cover monitoring in mountainous regions. The joint use of S2 and L8 greatly improves the mountainous snow cover monitoring ability.

In the 2021–2022 hydrological year, the overall snow cover area is relatively small, and the changes are more severe. Figure 7 depicts in detail the spatiotemporal distribution of snow cover in the BRB from September 2021 to August 2022, encompassing a total of 55 days of images. The bolded dates indicate snow cover extracted from cloud-covered images, while the red dates represent snow cover extracted from Landsat 8 images. The results suggest that, from December to January of the subsequent year, the snow cover area is relatively small. And frequent snowfall occurs in February and March, resulting in a higher proportion of snow cover area. In summer, the area of snow cover is smallest, and there is almost no snow in the watershed.

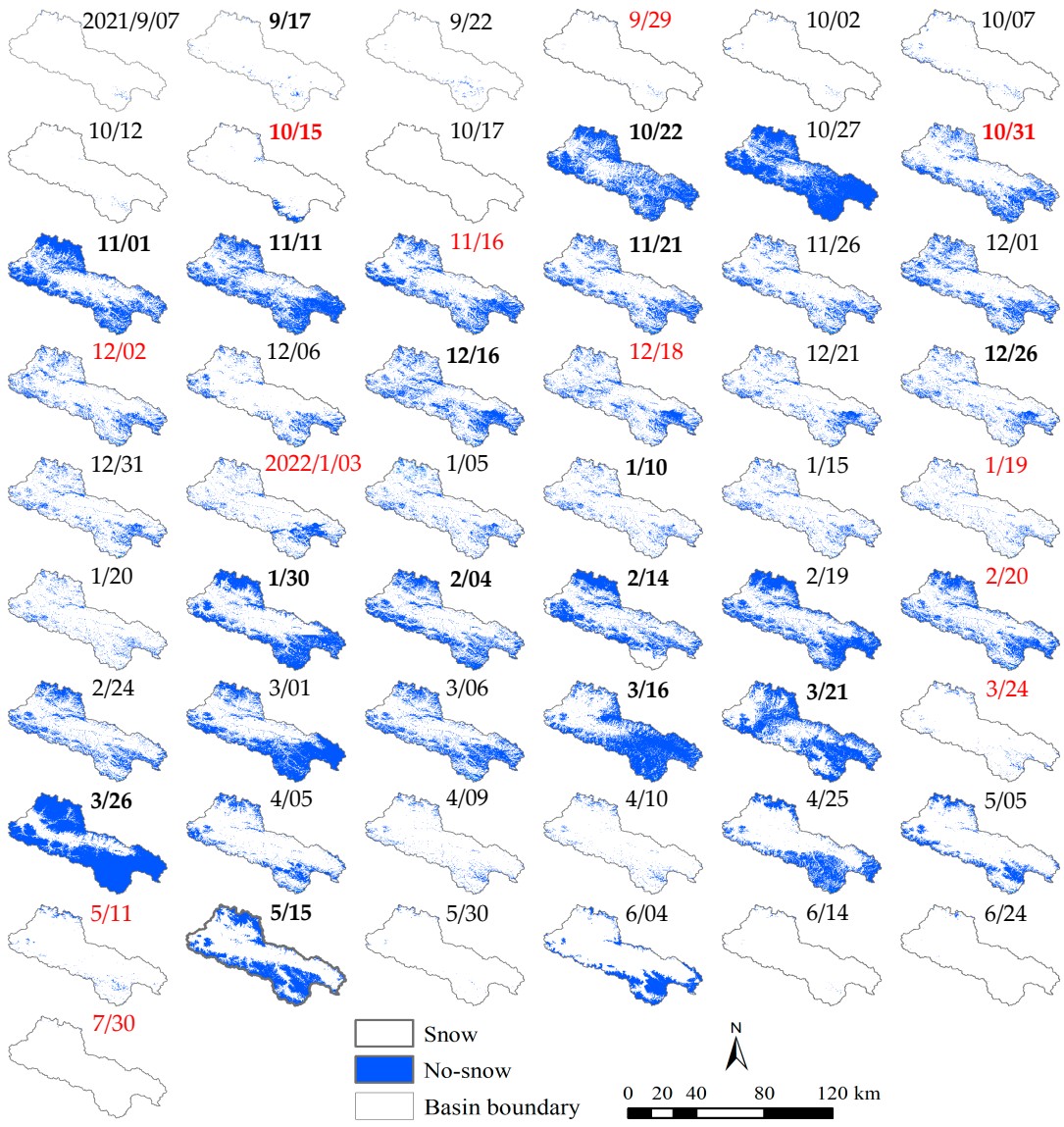

**Figure 7.** Spatiotemporal variations of snow cover from October 2021 to May 2022.

To further analyze the factors contributing to the fluctuating distribution of snow cover in Figure 7 mentioned above, the daily average temperature variations over the corresponding period in the watershed were obtained from the GEE ERA5-Land Daily dataset. Figure 8 illustrates a negative correlation between SCR and daily average temperature. SCR decreases with increasing temperature; otherwise, the opposite is true, which is consistent with our previous study [12].

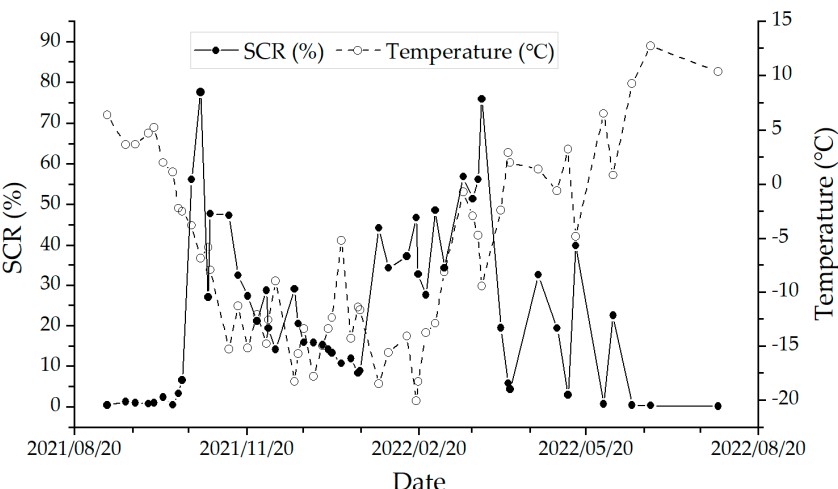

**Figure 8.** Relationship curve of SCR with temperature.

## 5. Discussion

The pollution of clouds and cloud shadows, coupled with the spectral differences between different satellite image bands, cause the comprehensive application of S2 and L8 to accurately monitor time-series snow cover variations to be of great significance in mountainous areas. Therefore, the following three aspects will be discussed: the advantages of integrating the two types of data, the inter-annual variation characteristics of snow cover in three hydrological years, and the limitations of the experiment.

### 5.1. Advantages of Combining Two Types of Satellite Data

In the Qilian Mountains, snow cover undergoes rapid changes, exhibiting strong spatiotemporal heterogeneity. In our published literature [12], 42 days of Sentinel-2 images were obtained to analyze the spatiotemporal distribution features of snow cover from September 2019 to August 2020. In this study, 11 days of Landsat 8 images were added, which can more comprehensively describe the time-series variation characteristics of snow cover. Figure 9 shows that Landsat 8 satellite imagery has increased the observation density and greatly improved the snow-cover-monitoring capability in mountain areas. As depicted within the red box in Figure 9, in the absence of Landsat 8 images during periods, the Sentinel 2 satellite alone would not be able to capture the actual changes in snow. Therefore, combining S2 and L8 images provides valuable insights for analyzing the time-series variation characteristics in mountainous snow cover.

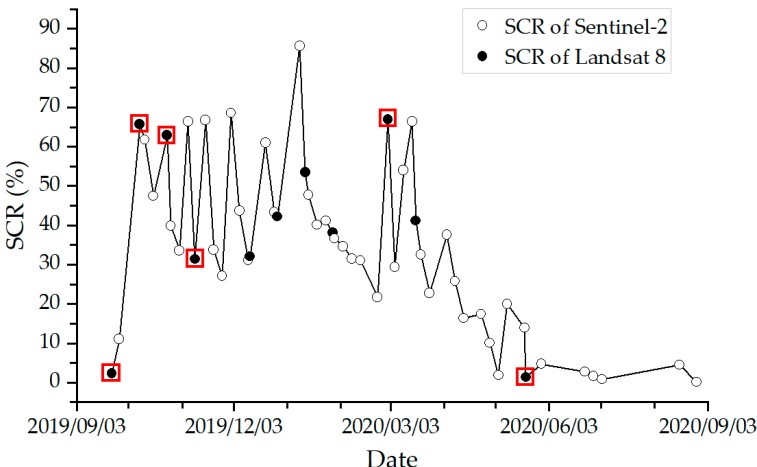

**Figure 9.** Time-series snow cover variation in the BRB by combining Sentinel-2 and Landsat 8 in the 2019–2020 hydrological year.

*5.2. Inter-Annual Variation Characteristics of Snow Cover in Three Hydrological Years*

As is well-known, the distribution of mountainous snow cover has strong spatial heterogeneity and rapid variations over time. Therefore, combining S2 and L8 images to reconstruct snow cover under CCSs can greatly improve the monitoring ability of snow cover variations. As shown in Figure 10, starting from early September, the snow cover area in BRB gradually increased, reaching its maximum in October; starting from March of the following year, it rapidly decreased and reached its minimum by mid-August. Comparing the fluctuation trend of snow cover area from 2019 to 2022, it is found that this is due to the influence of different periods of large-scale snowfall and the rapid melting of snow cover in mountainous areas. The snow cover area in BRB showed fluctuating changes within a year, while there were significant differences in the multi-peak morphology in different hydrological years. From September 2019 to August 2018, due to the combined influence of temperature and precipitation, large-scale snow accumulation mainly occurred in winter and spring. As summer temperatures rise, the snow begins to melt violently, but there are some periods of large-scale snowfall during this period. In the second hydrological year, the winter snow cover shows a trend of first increasing and then decreasing, while, in summer, it gradually decreases due to the influence of temperature. However, the winter snow cover in the third hydrological year was significantly smaller than in the first two years. From Figure 10, it can also be observed that there is a negative correlation between the trend of snow cover area changes and the daily average air temperature in the three hydrological years; that is, the snow cover area decreases with the increase of temperature.

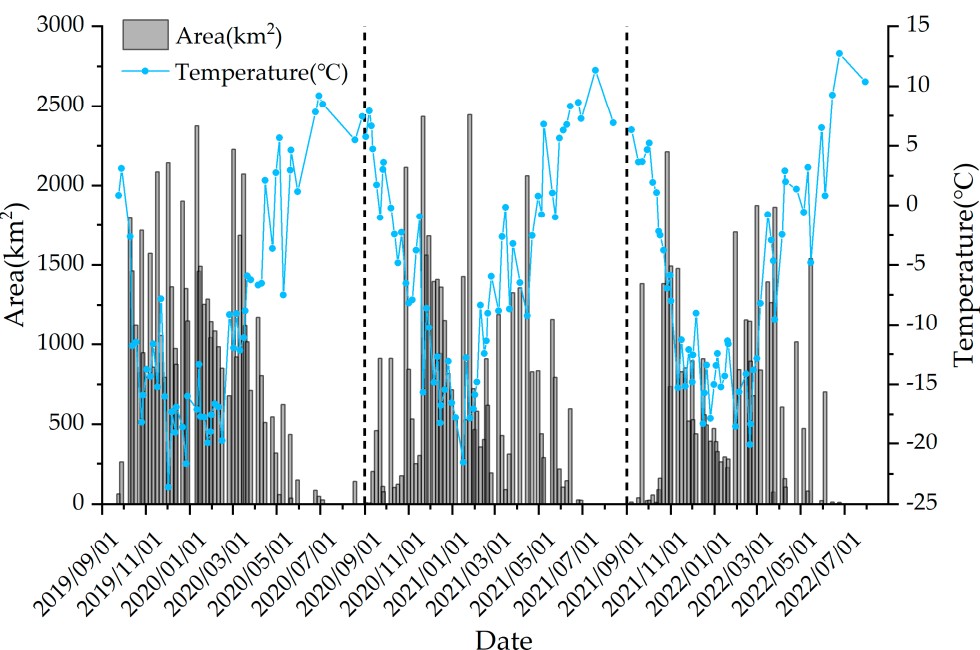

**Figure 10.** Inter-annual variation of snow cover area and the relationship with daily average air temperature in three hydrological years.

*5.3. Limitations of Experiments*

Although this study can effectively reconstruct snow cover under CCSs and greatly improve good snow detection capabilities in mountainous areas, it still has some limitations. First, the SNOWL algorithm does not consider the effects of factors such as solar radiation and wind on the distribution of snow cover. The intensity of wind can affect the distribution of snow, and the duration of solar radiation affects the rate of snow melting. Moreover, the improved cloud shadow detection method still uses a fixed threshold. The brightness and texture of cloud shadows vary with time and geographic location, and fixed thresholds often cannot capture this variation. Furthermore, there is a certain time difference between

the GF-2 used for accuracy verification and the transit times of S2 and L8 satellites. Due to the rapid changes in snow cover in mountainous areas, comparing the accuracy of GF-2 and S2 or L8 and S2 satellite data obtained on different days will reduce the accuracy and reliability of snow reconstruction.

In response to the above limitations, the next step of research will consider three factors: terrain, land-cover types underlying snow cover, and different snow cover periods to improve the fixed threshold for cloud shadow detection. Meanwhile, since the unmanned aerial vehicle (UAV) data are not affected by cloud interference and strong flexibility and maneuverability, they will be used as validation data. Additionally, due to the similar spectral characteristics between Landsat 9 and Landsat 8, it can also be incorporated into snow reconstruction applications. Therefore, combining three types of satellite remote-sensing images, namely, Sentinel-2, Landsat 9, and Landsat 8, will further enhance snow-cover-monitoring capabilities in mountainous areas.

## 6. Conclusions

This study uses different cloud detection methods for S2 and L8 and combines the improved Fmask4.0 cloud shadow detection algorithm, Sen2Cor, and our improved SNOWL algorithm published to reconstruct snow cover under CCSs. Two types of satellite imagery of 168 days were obtained in three hydrological years, and 45-day images covered by CCSs were successfully reconstructed. The experimental findings clearly demonstrate that the improved cloud shadow detection algorithm significantly enhances the accuracy of cloud shadow identification. Compared with the verification images, both satellite images can accurately recover snow cover covered by CCSs. The snow cover images of three hydrological years show significant differences in the spatiotemporal distribution of snow cover in BRB, with overall fluctuations and different periods of large-scale snowfall.

The specific innovations of this study are as follows: Firstly, the cloud shadow detection algorithm is improved by considering surface coverage types, which significantly enhances the accuracy of cloud shadow identification. Secondly, for S2 and L8 imagery, different methods are chosen to reconstruct snow cover under CCSs. Furthermore, an improved SNOWL algorithm is used to reconstruct snow cover under CCSs by removing unstable snow cover areas extracted from all S2 and L8 clear-sky images. Therefore, by integrating the above two types of images, snow cover variation information can be captured, which will greatly improve the ability to monitor the time-series variation in mountainous snow cover.

**Author Contributions:** Y.Z. designed the research and analyzed the article. C.Y. formulated the model and prepared original draft. R.Y. helped with the experiments and processed the satellite products. K.L. supervised the data processing and manuscript writing. All authors have read and agreed to the published version of the manuscript.

**Funding:** This research was funded the National Natural Science Foundation of China (Grant No. 42361058 and 41871277) and the Leading Talent Training Project of the Gansu Provincial Department of Natural Resources (Grant No. 202211).

**Data Availability Statement:** Data sharing is not applicable to this article.

**Acknowledgments:** Sentinel-2 A/B and Landsat 8, SRTM DEM, and ERA5-land Daily are all from the GEE data catalog (https://developers.google.com/earth-engine/datasets/, (accessed on 1 August 2023)). All authors are grateful to Jianguang Wen for providing the GF-2 data.

**Conflicts of Interest:** The authors declare no conflict of interest.

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
