# Peer review of "Reconstructing Snow Cover under Clouds and Cloud Shadows by Combining Sentinel-2 and Landsat 8 Images in a Mountainous Region"

_remotesensing, doi:10.3390/rs16010188_

Round 1

Reviewer 1 Report (Previous Reviewer 1)

Comments and Suggestions for Authors

The manuscript titled ‘Spatiotemporal reconstruction of snow cover under clouds and cloud shadows on Sentinel-2 and Landsat 8 time-series images in mountainous areas’ by Zhang et al. aims to combine S2 and L8 images for reconstructing snow cover in mountainous regions. However, there are several critical issues that need to be addressed before it can be considered for publication.

The title emphasizes ‘under cloud and cloud shadow’, yet this aspect is scarcely mentioned in the abstract, which instead emphasizes the improvement of capabilities in mountainous areas. Furthermore, the two innovations proposed in the abstract do not seem to relate to cloud and cloud shadow, leading to a discrepancy.

In the abstract, it is stated that ‘the overall accuracies for S2 and L8 are 80.74% and 88.81%...’. However, my understanding is that S2 and L8 are combined in the application, suggesting that only one accuracy should be achieved. This issue needs to be clearly illustrated.

The literature review is inadequate, and the shortcomings of existing studies are not well summarized. A comprehensive literature review is essential as it forms the foundation of the study objective.

At line 234, it is mentioned that ‘BT is retrieved from the ground temperature band of Landsat 8’. This seems to be a misunderstanding. BT, which stands for brightness temperature, is provided directly in Landsat 8 level 2 data, not from the ground temperature.

The manuscript's method of snow cover extraction is applied for both cloud-free and cloud-covered shadow conditions. It is not clear what the specific contribution or innovation of this manuscript is in this regard.

Sections 5.1 and 5.2 both belong to the results section and should be appropriately organized.

Author Response

Reviewer 1’s Comments and Our Responses

We sincerely thank you for your detailed, valuable, and insightful comments. We have carefully considered and revised your suggestions and have responded to your questions one by one. Your original comments were in black italics. Our response is in blue. Revisions are indicated by red text in the manuscript.

  1. The title emphasizes ‘under cloud and cloud shadow’, yet this aspect is scarcely mentioned in the abstract, which instead emphasizes the improvement of capabilities in mountainous areas. Furthermore, the two innovations proposed in the abstract do not seem to relate to cloud and cloud shadow, leading to a discrepancy.

Reply: We are very sorry that the meaning of this abstract is not clear. Thanks for your instruction, we have revised the abstract section.

Lines:16-20

Based on our previously published approach for snow reconstruction on S2 images using the Google Earth Engine (GEE), this study introduces two main innovations to reconstruct snow cover: (1) combining S2 and L8 images and choosing different CCSs detection methods and (2) improving the cloud shadow detection algorithm by considering land cover types, thus further improving the mountainous snow monitoring ability.

  1. In the abstract, it is stated that ‘the overall accuracies for S2 and L8 are 80.74% and 88.81%...’. However, my understanding is that S2 and L8 are combined in the application, suggesting that only one accuracy should be achieved. This issue needs to be clearly illustrated.

Reply: You are a careful expert. We will provide an explanation for your confusion. This article utilizes two types of satellite images with significant spectral differences. There is a significant difference in the methods of extracting snow cover beneath clouds and cloud shadows. And there may also be some differences in the accuracy of snow cover extraction, thus it is necessary to evaluate the accuracy separately.

  1. The literature review is inadequate, and the shortcomings of existing studies are not well summarized. A comprehensive literature review is essential as it forms the foundation of the study objective.

Reply: Thanks a lot for your question. In the previous draft, we had made careful revisions to the issue you raised (The literature review is not adequate and the shortcoming of the existed studies is not well summarized. This is the foundation of your study objective.). Meanwhile, based on your suggestion, we have rechecked the Introduction section and made modifications to the parts with logical issues.

Lines:85-89

Obviously, for S2 and L8 images with low temporal resolution and high spatial resolution, the SNOWL algorithm has become the main method for snow cover reconstruction under CCSs, and cloud and cloud shadow detection is a prerequisite. There are two main types of algorithms used for CCSs detection: machine learning and spectral threshold detection.

In order to facilitate your reading, we have attached the revised content reply section of the previous draft below. We hope that the revisions meet your approval.

Lines:43-48

In recent years, the successful launch of the Sentinel-2 (S2) and Landsat 8 (L8) satellites has brought new opportunities for mountainous snow cover identification and detection. Compared to the Landsat 1-7 series satellites, L8 has 12-bit radiometric resolution and reduced saturation for snow. However, due to its low revisit period (16 days), using L8 alone cannot capture the rapid changes in mountainous snow cover.

Lines:67-77

The temporal filtering method utilizes the short-term dynamic change in cloud layers to recover snow cover beneath the clouds, but it usually requires very high temporal resolution satellite images, such as revisiting cycles of more than 2 times in a day. The spatial filtering method typically uses adjacent cloud-free pixel information, but it ignores rapid changes in snow cover and is not suitable for mountainous regions with significant spatial heterogeneity [26]. The multisensor synthesis method integrates optical satellite images with microwave satellite images or ground observation data to enhance the recognition capability of snow cover under clouds [25,27]. However, the effectiveness of this method is limited by the low spatial resolution of passive microwave remote sensing data and the sparsity of ground observation stations in mountainous regions.

Lines:80-83

In addition, in previous MODIS snow products, the focus was on the impact of clouds on snow cover, neglecting the extraction of snow cover under cloud shadows. However, for high-resolution images, cloud shadows are also an important factor in extracting snow cover.

Lines:86-96

There are two main types of algorithms used for cloud and cloud shadow detection: machine learning and spectral threshold detection algorithms. Machine learning methods require many training samples and have difficulty achieving global universality [28-30]. Spectral threshold detection methods utilize the special physical features of clouds, such as brightness, whiteness and coolness, to construct multiple spectral indices, which are widely used [12,31-40]. For the problem of cloud pollution in Landsat series satellite, some scholars have proposed various CCSs algorithms based on spectral threshold detection methods, such as ACCA, Fmask, MCM, and LaSRC, through the differences in surface reflectance and temperature between clouds and ground objects [31,34-36]. The ACCA cloud detection algorithm targets Landsat 4/5 and Landsat 7 images, with high cloud omission and misalignment errors, and does not detect cloud shadows [32,33].

Lines:99-106

Candra et al. [35] developed the MCM algorithm for Landsat 8 to detect CCSs through the difference in reflectivity between cloud-covered and clear-sky pixels, but this is only effective for thick clouds and their shadows. LaSRC generates the cloud mask of L8 images during atmospheric correction and extracts cloud shadows by band thresholds, but the detection accuracy of thin clouds is low [41]. The above research indicates that the Fmask algorithm and its higher version can effectively detect CCSs information in Landsat series satellite images.

Lines:131-133

After the detection of CCSs, the SNOWL algorithm is applied to redefine cloud and cloud shadow pixels with elevations greater than the average snow line as snow cover under CCSs.

  1. At line 234, it is mentioned that ‘BT is retrieved from the ground temperature band of Landsat 8’. This seems to be a misunderstanding. BT, which stands for brightness temperature, is provided directly in Landsat 8 level 2 data, not from the ground temperature.

Reply: Very sorry, we have corrected several corresponding errors:

Lines:111-113

Because of the absence of brightness temperature (BT) observed in a cloud-sensitive thermal infrared band in S2 images

Lines:227-229

For Landsat 8, cloud pixels can be detected when the reflectivity value of the SWIR2 band is greater than 0.03 and the value of BT is less than 27 °C

Lines:233-234

 is reflectivity value of the SWIR2 band, and  represents the brightness temperature band of Landsat 8.

  1. The manuscript's method of snow cover extraction is applied for both cloud-free and cloud-covered shadow conditions. It is not clear what the specific contribution or innovation of this manuscript is in this regard.

Reply: Thank you for your question. Snow cover has strong spatiotemporal heterogeneity and fast spatiotemporal variation in mountainous areas. Only using S2 data cannot capture the rapid variations in snow cover, so L8 data was introduced. The high radiation resolution of S2 and L8 can reduce snow saturation. In addition, the combination of the two images can improve temporal resolution. However, the S2 and L8 images are severely affected by clouds and their shadows. Moreover, due to the spectral differences between the two images, the detection methods for clouds and cloud shadows are also different. Therefore, we transplanted all the algorithms in this study to GEE, and then studied the snow cover reconstruction under CCSs and the spatiotemporal variation characteristics of snow cover. This mainly including three aspects: (1) combining S2 and L8 images and choosing different CCSs detection methods; (2) improving the cloud shadow detection algorithm by considering land cover types, thus further improving the mountainous snow monitoring ability; (3) using an improved SNOWL algorithm to reconstruct snow cover under CCSs by removing unstable snow cover areas extracted from all S2 and L8 clear sky images, further improving the accuracy of snow cover reconstruction. I hope our explanation can answer your doubts.

  1. Sections 5.1 and 5.2 both belong to the results section and should be appropriately organized.

Reply: Thank you for your suggestion. Based on your suggestion, original sections 5.1 and 5.2 have been reorganized into section 4.4 of Results. In addition, two new sections have been added to Discussion, see sections 5.2 and 5.3 in the manuscript.

Lines:370-393

4.4. Impact of the terrain on accuracy of snow cover reconstruction

Terrain is an important factor affecting the reconstruction of snow cover in mountainous areas, with elevation and aspect being the most significant. According to the difference in elevation across the study area, the elevation zone is carried out with a step length of 200 m; thus, the S2 and L8 images are divided into five elevation zones. According to the reconstruction results of snow cover in Section 4.3, the cloud-free snow cover pixels on GF-2 and Sentinel-2 are used as true values to evaluate Sentinel-2 on March 21, 2020 and Landsat 8 on February 17, 2021 in different elevation zones, as shown in Table 5. The findings indicate significant variation in the overall accuracy of snow cover reconstruction among different elevation zones for Sentinel-2, but it re-mains above 83%, and the overall accuracy is better than that of Landsat 8. Although Table 5 shows a decreasing trend in overall accuracy as elevation increases, this varia-tion principle is unreliable due to the different SCRs in each elevation zone, especially at low altitudes.

Table 5. Impact of elevation on the reconstruction accuracy of snow cover.

Elevation

SCR

U

O

M

L

Unit: m

Unit: percentage(%)

S2

3474-3600

0.15

93.14

90.94

9.54

6.86

3600-3800

4.49

88.90

88.06

12.11

11.10

3800-4000

14.17

83.73

84.92

14.02

16.27

4000-4200

18.43

88.63

86.98

15.26

11.37

4200-4485

11.50

89.07

86.55

18.02

10.93

L8

3327-3600

1.22

63.38

84.83

13.50

36.62

3600-3800

4.57

64.73

85.81

5.37

35.27

3800-4000

9.02

82.30

90.74

4.93

17.70

4000-4200

13.08

89.65

91.84

5.85

10.35

4200-4434

7.41

85.41

89.82

5.83

14.59

To analyze the impact of aspect on the accuracy of snow cover reconstruction, it is categorized into five groups: flat surface (-1), shady slope (0-45°, 315-360°), semishady slope (45-135°), sunny slope (135-225°) and semisunny slope (225°-315°). As depicted in Table 6, the shady slope exhibits the highest overall accuracy for both S2 and L8, reaching 85.13% and 89.41%, respectively, followed by the semishady slope. In contrast, the overall accuracy on the sunny slope is the lowest because of the joint impact of the westerly wind, monsoon, and solar irradiance, which is 78.83% and 74.03% in S2 and L8, respectively. In addition, the SCR of aspects was not well correlated with the re-construction accuracy of snow cover in both satellite images.

Table 6. Impact of aspect on the reconstruction accuracy of snow cover.

Aspect(°)

S2

L8

SCR

U

O

M

L

SCR

U

O

M

L

Unit: percentage(%)

Flat surface

10.81

81.83

79.96

21.67

18.17

6.18

82.74

87.78

7.10

17.26

Shady slope

6.79

84.95

85.13

14.26

12.05

10.53

82.20

89.41

5.68

17.80

Semishady slope

10.84

77.95

81.86

14.38

22.05

6.54

85.41

84.74

16.57

14.59

Sunny slope

7.56

71.93

78.83

19.37

28.07

3.96

73.72

74.03

25.69

26.28

Semisunny slope

12.72

76.12

79.28

17.22

23.80

11.08

79.61

77.96

22.40

20.39

Lines:453-477

5.2. Interannual variation characteristics of snow cover in three hydrological years

Figure 10. Relationship between variations in snow cover area and daily average air temperature within three years.

As is well known, the distribution of mountainous snow cover has strong spatial heterogeneity and variations rapidly over time. Therefore, combining S2 and L8 imag-es to reconstruct snow cover under CCSs can greatly improve the monitoring ability of snow cover variations. As shown in Figure 10, starting from early September, the snow cover area in BRB gradually increased, reaching its maximum in October; starting from March of the following year, it rapidly decreased and reached its minimum by mid August. Comparing the fluctuation trend of snow cover area from 2019 to 2022, it is found that due to the influence of different periods of large-scale snowfall and the rapid melting of snow cover in mountainous areas. The snow cover area in BRB showed fluctuating changes within a year, while there were significant differences in the multi peak morphology in different hydrological years. From September 2019 to August 2018, due to the combined influence of temperature and precipitation, large-scale snow accumulation mainly occurred in winter and spring. As the rise of summer temperatures, the snow begins to melt violently, but there are some periods of large-scale snowfall during this period. In the second hydrological year, the winter snow cover shows a trend of first increasing and then decreasing, while in summer, it gradually decreases due to the influence of temperature. However, the winter snow cover in the third hydrological year was significantly smaller than in the first two years. From Figure 10, it can also be observed that there is a negative correlation be-tween the trend of snow cover area changes and the daily average air temperature in the three hydrological years, that is, the snow cover area decreases with the increase of temperature.

Lines:478-499

5.3. Limitations of experiments

Although this study can effectively reconstruct snow cover under CCSs and greatly improve good snow detection capabilities in mountainous areas, it still has some limitations. First, the SNOWL algorithm does not consider the effects of factors such as solar radiation and wind on the distribution of snow cover. The intensity of wind can affect the distribution of snow, and the duration of solar radiation affects the rate of snow melting. Moreover, the improved cloud shadow detection method still uses a fixed threshold. The brightness and texture of cloud shadows vary with time and geographic location, and fixed thresholds often cannot capture this variation. Furthermore, there is a certain time difference between the GF-2 used for accuracy verification and the transit times of S2 and L8 satellites. Due to the rapid changes in snow cover in mountains areas, comparing the accuracy of GF-2 and S2 or L8 and S2 satellite data obtained on different days will reduce the accuracy and reliability of snow reconstruction.

In response to the above limitations, the next step of research will consider three factors: terrain, land-cover types underlying snow cover, and different snow cover period to improve the fixed threshold for cloud shadow detection. Meanwhile, since the unmanned aerial vehicle (UAV) data is not affected by cloud interference and strong flexibility and maneuverability, it will be used as validation data. Additionally, due to the similar spectral characteristics between Landsat 9 and Landsat 8, it can also be incorporated into snow reconstruction applications. Therefore, combining three types of satellite remote sensing images, namely Sentinel-2, Landsat9 and Landsat8, will further enhance snow cover monitoring capabilities in mountainous areas.

Reviewer 2 Report (Previous Reviewer 3)

Comments and Suggestions for Authors

The author has made good changes or replies

Author Response

Thank you very much for your recognition of our paper and the revision work.

Reviewer 3 Report (New Reviewer)

Comments and Suggestions for Authors

This paper shows a spatiotemporal reconstruction of snow cover under clouds and cloud shadows by combining satellite images. The contribution is interesting. However, some points need to be better detailed for a complete understanding the article. Title: a lot of words. Focus on the main aspects. Keywords: ok. Abstract: Which country is the study area? Introduction: is comprehensive whit a good overview of problem in context.  Detail the reason for defining this area of study: Babao River Basin. Materials and Methods: the method description is good.  Considering it is a local experimental analysis, indicate the region/country in figure 1. Better describe the soils, relief, land use and land cover of the study region. It is important to include more details in the methodology, especially in the pre-processing of the L8 and S2 images used. Better justify the use GEE and of Fmask4.0 Algorithm. Results: is correctly interpreted. Indicate in figure 3, 4 and 5 which area covers the total study region. Discussions: part of the discussion text refers to results. The discussion should be more detailed and with more references to other studies on the topic (discussed results with only 1 study/author). They could better discuss and interpret the results from the perspective of previous studies and study hypotheses. For example: 1. It would be interesting to detail possible other influences (or not) of different levels of processing or time of years (September 2019 to August 2022). 2. Comment further on possible limitations and challenges of the GEE and of Fmask4.0, Algorithm SNOWL Algorithm, … 3. The study's applications for land use and cover analysis. 4. Comments further on sources of uncertainty and accuracy of this analysis relative to other analyzes on the topic. Conclusions: The initial part could go to the Discussion item. The text should focus more on the main results and highlight’s. Need to improve the formatting of the text. Examples: Introduction “snow cover”, Table 1., 2.2.1., 2.2.2, 3.1.2… (spacing), Figure 2. (cut letters), Figure 3 and 4 (position), Figure 7. spatiotemporal,

Author Response

Reviewer 3’s Comments and Our Responses

We sincerely thank you for your detailed, valuable, and insightful comments. We have carefully considered and revised your suggestions and have responded to your questions one by one. Your original comments were in black italics. Our response is in blue. Revisions are indicated by red text in the manuscript.

  1. Title: a lot of words. Focus on the main aspects.

Reply: Thank you for your suggestion. The title has been modified to " Reconstructing snow cover under clouds and cloud shadows by combining Sentinel-2 and Landsat 8 images in a mountainous".

  1. Abstract: Which country is the study area?

Reply: The research area is located in the northeast of the Qilian Mountains in China. We have added country information in Abstract.

Lines:20-21

The Babao River Basin of the Qilian Mountains in China is chosen as the study area

  1. Detail the reason for defining this area of study: Babao River Basin.

Reply: Thank you very much for your suggestion. We have added the reason for choosing BRB as the research area in section 2.1, as explained below,

Lines:174-177

There are important snow observation sites within the watershed that are convenient for snow monitoring, such as the Dadongshuyakou Observation Station. Due to its unique geographical location and hydrological characteristics, the BRB has become a desired area for researching snow cover variation in cold regions.

  1. Materials and Methods: the method description is good.Considering it is a local experimental analysis, indicate the region/country in figure 1. Better describe the soils, relief, land use and land cover of the study region.

Reply: According to your suggestion, we have added country information to Figure 1 and provided a detailed description of land use within the study area.

Figure 1. Location of the study area.

Lines:169-171

The dominant land cover types are grassland (77.20%), bare land (11.14%), moss (8.61%), and alpine woodland (1.76%) [54,55].

  1. It is important to include more details in the methodology, especially in the pre-processing of the L8 and S2 images used. Better justify the use GEE and of Fmask4.0 Algorithm.

Reply: Thank you for your suggestion. In Methodology, we have added the processing procedures for two types of satellite images and the rationality of using the Fmask4.0 algorithm on the GEE platform.

Lines:209-212

This study used JavaScript language to process S2 and L8 data on the GEE platform, and modified the Python version of Fmask4.0 algorithm to a JavaScript version supported by GEE. Additionally, Both Sen2Cor and SNOWL algorithms were also written in JavaScript. And all data were resampled to a spatial resolution of 10 meters for subsequent snow extraction.

  1. Results: is correctly interpreted. Indicate in figure 3, 4 and 5 which area covers the total study region.

Reply: Thank you for pointing out the issue. However, in order to reduce the length of the article, the corresponding sample area locations were not provided in the figures. To address your confusion, we have created a location map showing the positions of the sample areas within the study area. For example, Figure 3 (a), Figure 4 (e), and Figure 5 (d) depict the respective sample area locations.

  1. Discussions: part of the discussion text refers to results. The discussion should be more detailed and with more references to other studies on the topic (discussed results with only 1 study/author). They could better discuss and interpret the results from the perspective of previous studies and study hypotheses. For example:

1) It would be interesting to detail possible other influences (or not) of different levels of processing or time of years (September 2019 to August 2022).

2) Comment further on possible limitations and challenges of the GEE and of Fmask4.0, Algorithm SNOWL Algorithm, …

3) The study's applications for land use and cover analysis.

4) Comments further on sources of uncertainty and accuracy of this analysis relative to other analyzes on the topic.

Reply: Thank you very much for your detailed suggestions. The discussion has been supplemented in detail based on your suggestions, with supplementary content in sections 5.2 and 5.3. Additionally, due to the limited research on snow cover reconstruction of high-resolution images under clouds and cloud shadows in BRB, this paper did not compare with existing literature. However, we will focus on related research in the future.

Lines:432-437

The pollution of clouds and cloud shadows, coupled with the spectral differences between different satellite image bands, make the comprehensive application of S2 and L8 to accurately monitor time-series snow cover variations a great significance in mountainous areas. Therefore, the following three aspects will be discussed: the ad-vantages of integrating the two types of data, the interannual variation characteristics of snow cover in three hydrological years, and the limitations of the experiment.

Lines:453-477

5.2. Interannual variation characteristics of snow cover in three hydrological years

Figure 10. Interannual variation of snow cover area and the relationship with daily average air temperature in three hydrological years.

As is well known, the distribution of mountainous snow cover has strong spatial heterogeneity and variations rapidly over time. Therefore, combining S2 and L8 images to reconstruct snow cover under CCSs can greatly improve the monitoring ability of snow cover variations. As shown in Figure 10, starting from early September, the snow cover area in BRB gradually increased, reaching its maximum in October; starting from March of the following year, it rapidly decreased and reached its minimum by mid August. Comparing the fluctuation trend of snow cover area from 2019 to 2022, it is found that due to the influence of different periods of large-scale snowfall and the rapid melting of snow cover in mountainous areas. The snow cover area in BRB showed fluctuating changes within a year, while there were significant differences in the multi peak morphology in different hydrological years. From September 2019 to August 2018, due to the combined influence of temperature and precipitation, large-scale snow accumulation mainly occurred in winter and spring. As the rise of summer temperatures, the snow begins to melt violently, but there are some periods of large-scale snowfall during this period. In the second hydrological year, the winter snow cover shows a trend of first increasing and then decreasing, while in summer, it gradually decreases due to the influence of temperature. However, the winter snow cover in the third hydrological year was significantly smaller than in the first two years. From Figure 10, it can also be observed that there is a negative correlation between the trend of snow cover area changes and the daily average air temperature in the three hydrological years, that is, the snow cover area decreases with the increase of temperature.

Lines:478-499

5.3. Limitations of experiments

Although this study can effectively reconstruct snow cover under CCSs and greatly improve good snow detection capabilities in mountainous areas, it still has some limitations. First, the SNOWL algorithm does not consider the effects of factors such as solar radiation and wind on the distribution of snow cover. The intensity of wind can affect the distribution of snow, and the duration of solar radiation affects the rate of snow melting. Moreover, the improved cloud shadow detection method still uses a fixed threshold. The brightness and texture of cloud shadows vary with time and geographic location, and fixed thresholds often cannot capture this variation. Furthermore, there is a certain time difference between the GF-2 used for accuracy verification and the transit times of S2 and L8 satellites. Due to the rapid changes in snow cover in mountains areas, comparing the accuracy of GF-2 and S2 or L8 and S2 satellite data obtained on different days will reduce the accuracy and reliability of snow reconstruction.

In response to the above limitations, the next step of research will consider three factors: terrain, land-cover types underlying snow cover, and different snow cover period to improve the fixed threshold for cloud shadow detection. Meanwhile, since the unmanned aerial vehicle (UAV) data is not affected by cloud interference and strong flexibility and maneuverability, it will be used as validation data. Additionally, due to the similar spectral characteristics between Landsat 9 and Landsat 8, it can also be incorporated into snow reconstruction applications. Therefore, combining three types of satellite remote sensing images, namely Sentinel-2, Landsat9 and Landsat8, will further enhance snow cover monitoring capabilities in mountainous areas.

  1. Conclusions: The initial part could go to the Discussion item. The text should focus more on the main results and highlight’s.

Reply: Sincerely thank you for your suggestion. According to your suggestion, the first part of Conclusion has been included in Discussion, see lines 432-435. In addition, the conclusion section has reorganized the research focus of the manuscript, please refer to the "Conclusion" for details.

Lines:432-435

The pollution of clouds and cloud shadows, coupled with the spectral differences between different satellite image bands, make the comprehensive application of S2 and L8 to accurately monitor time-series snow cover variations a great significance in mountainous areas.

Lines:508-520

The snow cover images of three hydrological years show significant differences in the spatiotemporal distribution of snow cover in BRB, with overall fluctuations and different periods of large-scale snowfall.

The specific innovations of this study are as follows. Firstly, the cloud shadow detection algorithm is improved by considering surface coverage types, which significantly enhances the accuracy of cloud shadow identification. Secondly, for S2 and L8 imagery, different methods are chosen to reconstruct snow cover under CCSs. Furthermore, an improved SNOWL algorithm is used to reconstruct snow cover under CCSs by removing unstable snow cover areas extracted from all S2 and L8 clear sky images. Therefore, by integrating the above two types of images, snow cover variation information can be captured, which will greatly improve the ability to monitor time-series variation in mountainous snow cover.

  1. Need to improve the formatting of the text. Examples: Introduction “snow cover”, Table 1., 2.2.1., 2.2.2, 3.1.2… (spacing), Figure 2. (cut letters), Figure 3 and 4 (position), Figure 7. spatiotemporal,

Reply: Thank you very much for your careful review. We have carefully reviewed all the formatting of the text, including: spacing, cut letters, and position. For example,

Figure 1. Location of the study area.

Figure 2. Flowchart of this algorithm.

Figure 7. Spatiotemporal variations of snow cover from October 2021 to May 2022.

Reviewer 4 Report (New Reviewer)

Comments and Suggestions for Authors

Thank you, this manuscript looked much-improved over the previous version.  I appreciate the changes you made to the annotations on the figures. I am happy to accept this in its present form. 

Author Response

Thank you very much for your recognition of our paper and the revision work.

Round 2

Reviewer 1 Report (Previous Reviewer 1)

Comments and Suggestions for Authors

No further questions.

Reviewer 3 Report (New Reviewer)

Comments and Suggestions for Authors

The authors made the suggested adjustmentes.

This manuscript is a resubmission of an earlier submission. The following is a list of the peer review reports and author responses from that submission.

Round 1

Reviewer 1 Report

Comments and Suggestions for Authors

The manuscript titled ‘Spatiotemporal reconstruction of snow cover under clouds and cloud shadows on Sentinel-2 and Landsat 8 time-series images in mountainous area’ by Zhang et al. combines S2 and L8 images and improves the Fmask4.0 cloud shadow detection algorithm to further improve the monitoring ability of snow cover changes in mountainous areas. There are some critical issues that should be paid attention to before it can be accepted for publish.

The title focuses on ‘under cloud and cloud shadow’, while in your abstract this part is rarely mentioned and it emphasized that ‘… improve the ability in mountainous areas’. This discrepancy confused me. What’s your main objective, to deal with cloud and cloud shadow or mountainous effect in snow cover reconstruction?

The innovation of adopted method should be clearly stated in your abstract.

The literature review is not adequate and the shortcoming of the existed studies is not well summarized. This is the foundation of your study objective.

At line 204, the criteria for cloud detection is BT < 27℃, while at line 208 it’s the ground temperature. Which one are you used? Besides, it’s suggested to use Kelvin temperature instead of Celsius temperature in a scientific manuscript.

Your improved cloud shadow detection method, especially formula 6 and 7, should be clarified. Why this kind of criteria and threshold are adopted?

Your published method of snow cover extraction is adopted for both cloud free and CCS condition, so what’s the contribution or innovation of this manuscript?

I’m especially curious about the result of your reconstructed snow cover map, in other words a comparison of original snow cover and your reconstructed snow cover. This is your main target. However, I do not find it in your result. 

Section 5.1 and 5.2 both belong to result section.

Comments on the Quality of English Language

The language is rough and there are many language issues in the text. Many grammar errors and inconsistent tense exist here and there. It’s suggested to check the whole manuscript and polish the language.

Reviewer 2 Report

Comments and Suggestions for Authors

Please revise and resubmit:
Some general comments to consider. 

I'd like to see the qualification of the pollution of clouds and cloud shadows expanded a bit more in the manuscript to consider cloud layering and any impacts from vertically developed clouds.  Perhaps an assessment of performance between cumulus and stratus clouds would suffice? 

Watch you use of the words "accurately" and "properly" -- there some be some careful quantification of how accurately (confidence degree etc).  

Snow cover and snow properties are a key element to determining how well your satellite-based data processing algorithms will perform.  Have you compared your algorithm across different snow densities, morphologies, and seasonal evolution of snow-surfaces?   This seems to be a somewhat large shortcoming of this paper, but perhaps there are some ground-based snow cover "snow pit" datasets that could provide necessary insight?   

I see a number of edits have been made to the text to add words, clarifications etc.  I am left wondering if the paper has had a careful review for grammar, proof-reading etc, and final english-style check.  As always, please check prior to resubmission. 

Comments on the Quality of English Language

All the red text has left me thinking the paper needs one more careful proof read on resubmission, which you would do anyway. 

Reviewer 3 Report

Comments and Suggestions for Authors

This study used their comprehensive method published to reconstruct snow cover covered by clouds and shadows on Sentinel-2 and Landsat 8 imagery. They used Fmask to mask the clouds and cloud shadows, Sen2Cor to map the snow covered areas for clear sky conditions and snow-line algorithm which depends on DEM to improve the snow mapping under clouds and cloud shadows. They examines 399 scenes of Sentinel-2 images and 35 scenes of Landsat 8 images for the Babao River Basin from September 2019 to August 2022. The results show that the proposed method can improve the mountainous snow cover detection ability.

Overall, the study is interesting and within the scope of the journal. However, there are some problems in the manuscript that can be addressed before publication.

Specific comments:

1.        Many part of the paper are marked with red text. Please check if the submitted paper is the final manuscript.

2.        Lines 170-171: Sentinel-2 L2A and Landsat 8 SR are the surface reflectance data, which are all secondary products, while their cirrus bands are primary products (Sentinel-2 L1C and Landsat 8 TOA). I am very confused about the necessity of introducing the cirrus band and image quality issues through calculations between different levels of images.

3.        Line 183: The manuscript mentions that the "Meanvis" algorithm is used to extract snow cover from GF-2. How is the threshold determined? Is this threshold applicable in other regions?

4.        Figure 3 and 5: How did you use GF-2 imagery (0.8 m) for accuracy verification, as there are significant differences in spatial resolution among the three types of satellite images?

5.        The geometric position relationship between clouds and cloud shadows can effectively determine the position of cloud shadows. Why did the manuscript only use the method of band threshold without considering the joint application of the two?

6.        If the cloud shadow detection results can be displayed under different underlying land cover types, it will be more interesting, and it can determine whether the shadows in the image are cloud shadow or water.

7.        In Figures 5 (a) and (d), the Babao River Basin is in the freezing period, and the rivers in the figure should be covered by ice. Since this study focuses on extracting snow cover, how does the algorithm distinguish between ice and snow?

8.        There are some formatting errors, please review and correct them throughout:

a)        Table1: Please unify the date format and check it throughout.

b)        Table6: The header should be shown in bolded.

9.        Line 328: The manuscript mentioned that " in areas with dense cloud cover, the accuracy of snow cover reconstruction is low...". However, the paper has never explicitly stated the requirements of the SNOWL algorithm for cloud coverage in images, whether it is possible to perform snow cover reconstruction under clouds after the image is completely covered by clouds?

10.    Line 386: Is' Table 7 'a writing error? I think it should be ' Table 5'.

11.    In addition, some important references of spatiotemporal variability of snow cover China are not mentioned in the paper.